# A Highly Sensitive SERS and RRS Coupled Di-Mode Method for CO Detection Using Nanogolds as Catalysts and Bifunctional Probes

**DOI:** 10.3390/nano10030450

**Published:** 2020-03-02

**Authors:** Dongmei Yao, Guiqing Wen, Lingbo Gong, Chongning Li, Aihui Liang, Zhiliang Jiang

**Affiliations:** 1Key Laboratory of Ecology of Rare and Endangered Species and Environmental Protection (Guangxi Normal University), Ministry of Education, Guilin 541004, China; dmyao47@163.com (D.Y.); gqwen@mailbox.gxnu.edu.cn (G.W.); zljiang89@126.com (L.G.); lcn7882342@163.com (C.L.); 2Guangxi Key Laboratory of Environmental Pollution Control Theory and Technology for Science and Education Combined with Science and Technology Innovation Base, Guilin 541004, China; 3College of Chemistry and Biology Engineering, Hechi University, Yizhou 546300, China

**Keywords:** CO, nanocatalysis, SERS, RRS, coupled di-mode

## Abstract

Carbon monoxide (CO) is a commonly poisonous gas. It is important to detect CO in daily life. Herein, a new and sensitive surface enhanced Raman scattering (SERS) and resonance Rayleigh scattering (RRS) coupled di-mode method was developed for CO, based on gold nano-enzyme catalysis and gold nanoprobes. CO can react with HAuCl_4_ to generate gold nanoparticles (AuNPs) in pH 5.2 HAc-NaAc buffer. The generated AuNPs exhibited SERS activity at 1620 cm^−1^ in the presence of Vitoria blue B (VBB) molecular probes, and an RRS peak at 290 nm. Based on the AuNP bifunctional probes, the increased SERS and RRS intensities respond linearly with the concentration of CO in the range of 100–1500 ng/mL and 30–5230 ng/mL, respectively. To improve the sensitivity, the produced AuNPs were used as nano-enzyme catalysts for the new indicator reaction of HAuCl_4_-ethanol (En) to amplify the signal. The sensitive SERS method was coupled with the accurate RRS method to develop a sensitive and accurate SERS/RRS di-mode method for determination of 3.0–413 ng/mL CO, based on the AuNP-HAuCl_4_-En nanocatalytic reaction and its product of AuNPs as SERS and RRS bifunctional probes.

## 1. Introduction

The progress and continuity of metabolic and genetic information transmission in life-sustaining activities are closely related to the orderly chemical reactions in vivo, and enzymes are the key to catalyzing these chemical reactions to proceed smoothly. Bio-enzymes have the advantages of high catalytic efficiency, selectivity of substrates and wide range of catalytic reactions, but their application in production practice is greatly limited because of their high production cost, high cost of preservation and transportation, harsh use conditions and narrow scope [1,2,3,4]. Since the discovery of Fe_3_O_4_ nanomaterials with catalytic properties similar to horseradish peroxidase [5], nano-enzymes have attracted much attention. Nano-enzymes have the characteristics of high catalytic efficiency, stability, economy and large-scale preparation. They have been applied in medicine, chemical industry, food, agriculture, environment and analytical chemistry. The catalytic activity of nano-enzymes is higher than that of many traditional mimetic enzymes. Traditional mimetic enzymes are mostly based on organic complexes, using the molecular structure and microenvironment of mimetic enzymes’ catalytic centers such as polymer micelles or molecular imprinting technology. The common problem of these mimetic enzymes is that their catalytic activity and selectivity are low, and their preparation cost is high. So far, these mimetic enzymes can not meet the actual needs. However, nano-enzymes have similar catalytic activities to natural enzymes. The most important feature of nano-enzymes is that they have unique physical and chemical properties of nanomaterials besides catalytic functions. Therefore, nano-enzymes are also considered as bifunctional or multifunctional molecules. How to skillfully combine the catalytic activity of nano-enzymes with their physical and chemical properties to create more unique new functions will be a new subject to be studied. In analytical chemistry, nano-enzymatic reactions have been used in spectrophotometry, fluorescence and chemiluminescence [6,7,8]. Jiang et al. found that gold nanoparticles have catalase-like activity and can catalyze the oxidation of 3,3,5,5-tetramethylbenzidine (TMB) by H_2_O_2_ to produce a blue product [9]. Coupled glucose peroxidase (GOx) and gold nanoparticles can be used for the spectrophotometric determination of glucose content, with a linear range of 18–1100 μmol/L, and a detection limit of 4 μmol/L. Liu et al. prepared platinum nanodots in situ on gold nanorods. The nanocomposite could catalytically oxidize o-phenylenediamine (OPD) to produe yellow product, and the coupling GOx could detect glucose spectrophotometrically [10]. Based on the catalysis of immune nanoprobes and aptamer nanoprobes, they are also used in resonance Rayleigh scattering (RRS) and surface enhanced Raman scattering (SERS) quantitative analyses, respectively [11,12].

SERS is a sensitive molecular spectral detection technique based on nanosurface plasmon resonance including solid nanosurface and nanosol [13,14,15,16]. SERS works have not only been focused on the qualitative analysis but also on the quantitative analysis, especially, the improvement of its accuracy. To improve precision, some procedures were reported, such as the internal standard method and SERS/fluorescence di-mode method [12,17]. RRS uses molecular scattering, which can occur when the particle size is much smaller than the wavelength of incident light. The diameter of gold or silver nanoparticles is usually less than 50 nm, which is much smaller than the wavelength of the incident light. Therefore, gold or silver nanoparticles are often used as carriers for the RRS phenomenon. RRS is a simple, rapid and sensitive spectral analysis technology, which was used in nucleic acids, proteins, and small molecule analyses [18,19,20,21,22,23,24]. However, there are rare reports about the detection of gas pollutants by RRS. Wen et al. [25] established a rapid RRS method for 0.2–50 µmol/L O_3_, based on the generation of tri-iodine ion (I_3_^−^) reacting with Victoria blue B (VBB) to form the associated particle (VBB-I_3_)_n_ with an RRS peak at 722 nm_._ To the best of our knowledge, there are no reports about coupling SERS with RRS to set up a SERS/RRS di-mode method for quantitative analysis of CO, based on the produced AuNP catalysis of the HAuCl_4_-En-VBB system.

CO is a kind of colorless, odorless, poisonous gas, which is mainly generated by automobiles, incomplete combustion of fossil fuels, industrial production and residential environments [26,27]. People with chronic poisoning of carbon monoxide pollution cannot be aware of it completely, sometimes even feeling comfortable, which makes carbon monoxide more harmful. The chemical affinity of CO with hemoglobin is 300 times stronger than that of O_2,_ and considering its toxicity and difficulty to be detected, CO is regarded a very dangerous “silent killer” [28]. Many researchers have studied how to quickly and accurately detect CO, most studies focus on new sensors. Kim et al. [29] developed a Pd/SnO_2_ sensors that can detect CO below 60 °C. Boehm et al. [30] produced a device based on InP and GaSb. This device has two ways of the continuous wave mode to detect CO, which makes it possible for the lower limit of detection to reach 2.0 µg/mL. Wang et al. [31] studied the reactions of CO and NO in single wall boron nitride nanotubes (BNNTs) mixed with Ge. They showed that the performance of BNNTs was improved for CO and NO detection. Ghosh et al. [32] introduced a kind of stable and economically viable sensor based on SnO_2_ nanocrystals. It can achieve determination of CO in the air, with a detection limit of 1.0 μg/mL. Chen et al. [33] used bovine hemoglobin with a high affinity for CO to combine dissolved CO in water to form a two-component system consisting of hemoglobin (Hb) and carboxyhemoglobin (HbCO). CO content can be detected quantitatively using a tilapia color pool through fast and simple double wavelength ultraviolet visible spectrophotometry, of which the detection range achieved 0–0.15 μg/mL and the detection limit was 0.23 μg/L. Chen et al. [34] reported a sensor using a quantum cascade laser with an excitation wavelength of 4.65 µm as the light source, with a detection limit of 108 ppb. Zoltan et al. [35] studied a method which can detect CO concentration and total pressure of the light bulb plenum chamber in order to control the chemical purity of molybdenum foil indirectly through traditional Fourier transform infrared spectroscopy. Wang et al. [36] established a quick and easy method of CO detection for aquatic products, in which the detection limit reached 10 µg/kg. Wang et al. [37] prepared a near-infrared fluorescent probe for the determination of CO, based on the transformation of a nitro group into an amino moiety under the direct reduction of CO. Hao et al. [38] established a kind of headspace gas chromatography–mass spectrometry (HS/GC/MS) method for the precise determination of trace CO in rotten blood, which provides an accurate and reliable method for the determination of CO. While the methods mentioned above are mostly semi-quantitative, establishing a high sensitivity, good selectivity, simple operation and rapid detection method for CO is of great significance to environmental protection and human health. This paper studied the nano-catalytic reaction of CO-HAuCl_4_-En, and established a new SERS/RRS coupled di-mode method for trace CO. Compared with the reported method of CO, this method has advantages of accuracy and high sensitivity.

## 2. Materials and Methods

### 2.1. Apparatus

A model of a DXR smart Raman spectrometer (Thermo Fisher Scientific, Waltham, MA, USA) with laser of 633 nm, power of 2.5 mW, collect time of 5 s and slit of 25 μm, a model of F-7000 Hitachi fluorescence spectrometer (Hitachi Company, Tokyo, Japan) with volt = 500 V, excited slit = emission slit = 5 nm, emission filter = 1%T attenuator and λ_ex_ − λ_em_ = Δλ = 0, and a model of TU-1901 double beam UV-Vis spectrophotometer (Beijing Purkinje General Instrument Co., Ltd., Beijing, China) were used. A model S-4800 scanning electron microscope (SEM, Hitachi High-Technologies Corporation, Tokyo, Japan) was used to record the SEM images. A model of JEM-800 H Field emission transmission electron microscope (Hitachi High-Technologies Corporation, Tokyo, Japan) was used to record the transmission electron microscope (TEM) and energy spectrum, with lattice resolution of 0.204 nm, dot resolution of 0.45 nm, acceleration voltage of 200 kV and tilt angle of 25 degrees. A model of C-MAG HS7 Heating magnetic stirrer (IKA Company, Berlin, Germany), Constant temperature magnetic stirrer (Beijing Kewei Yongxing Instrument Company, Ltd., Beijing, China), and KC-6120 Atmospheric sampler (Laoshan Mountain Electronic Instrument Factory Company, Ltd., Laoshan, China) were used.

### 2.2. Reagents

A total of 20 mL water in a 25 ℃ water bath in a 25 mL tube was used and filled carbon monoxide gas into the water for 10 min under the normal atmospheric pressure, in which the concentration of CO was 26.03 mg/L according to reference [39]. A 0.2 mol/L HAc-NaAc (Shanghai Reagent Factory, Shanghai, China) buffer solution (pH = 5.2) was made by mixing 79 mL 0.2 mol/L NaAc solution with 21 mL 0.3 mol/L acetic solution then diluting to 100 mL. A 1.0 mg/mL HAuCl_4_ (Tianjin Guangfu Fine Chemical Research Institute, Tianjin, China) and 20 µmol/L Vitoria blue B (VBB, China Pharmaceutical Shanghai Chemical Reagent Station, Shanghai, China) solution was prepared. Ethanol, PdCl_2_, glucose and trisodium citrate were purchased from Shanghai Reagent Factory (Shanghai, China).

### 2.3. Experiment Methods

**CO-HAuCl_4_ system:** In a 5 mL empty test tube, 200 µL pH 5.2 HAc-NaAc buffer, 100 µL 0.2 mg/mL HAuCl_4_ and 120 µL 26.03 mg/L CO were added and mixed well. After 5 min, 57 µL 20 µmol/L VBB and 30 µL 1 mol/L NaCl solutions were added, shaken well, and then diluted to 1.5 mL with water. The SERS and RRS spectra were recorded through a Raman spectrometer and fluorescence spectrophotometer, respectively. The SERS at 1620 cm^−1^ (I) and RRS intensity at 290 nm (I_290 nm_), and the blank [I_0_, (I_290 nm_)_0_] without of CO were recorded. The ΔI = I − I_0_ and ΔI_290 nm_ = I_290 nm_ − (I_290 nm_)_0_ were calculated.

**CO-HAuCl_4_-ethanol catalytic system:** In a 5 mL empty test tube, 200 µL pH 5.2 HAc-NaAc buffer, 100 µL 0.2 mg/mL HAuCl_4_ and 120 µL 26.03 mg/L CO were added, diluted to 0.8 mL with water, and mixed well. After 5 min, a 200 µL 99.7% ethanol, 110 µL 0.01 mol/L HCl and 80 µL 1.0 mg/mL HAuCl_4_ were added. The tube was heated to 75 °C in a water bath for 20 min, and cooled with ice–water. Then, 57 µL 20 µmol/L VBB and 30 µL 1 mol/L NaCl solutions were added, and diluted to 1.5 mL with water. The SERS at 1620 cm^−1^ (I) and RRS intensity at 370 nm (I_370 nm_), and the blank [I_0_, (I_370 nm_)_0_] without of CO were recorded. The ΔI = I − I_0_, ΔI_370 nm_ = I_370 nm_ − (I_370 nm_)_0_ and ΔI_Di-mode_ = ΔI + ΔI_370 nm_ were calculated.

**CO-PdCl_2_ system:** In a 5 mL empty test tube, 200 µL pH 5.2 HAc-NaAc buffer, 150 µL 0.1 mg/mL PdCl_2_ and 120 µL 26.03 mg/L CO were added, mixed well, and then diluted to 1.5 mL with water. After 5 min, The RRS spectra were recorded through a fluorescence spectrophotometer. The RRS intensity at 370 nm (I), and the blank (I_0_) without of CO were recorded. The ΔI = I − I_0_ were calculated.

**CO-HAuCl_4_-glucose catalytic system:** In a 5 mL empty test tube, 200 µL pH 5.2 HAc-NaAc buffer, 100 µL 0.2 mg/mL HAuCl_4_ and 120 µL 26.03 mg/L CO were added, diluted to 0.8 mL with water, and mixed well. After 5 min, 100 µL 0.1 mol/L glucose, 100 µL 0.01 mol/L HCl and 80 µL 1.0 mg/mL HAuCl_4_ were added. The tube was heated in a 75 °C water bath for 20 min, and cooled with ice–water. Then, 57 µL 20 µmol/L VBB and 30 µL 1 mol/L NaCl solutions were added, and diluted to 1.5 mL with water. The SERS at 1620 cm^−1^ (I), and the blank (I_0_) without of CO were recorded. The ΔI = I − I_0_ were calculated.

**CO-HAuCl_4_-citrate catalytic system:** In a 5 mL empty test tube, 200 µL pH 5.2 HAc-NaAc buffer, 100 µL 0.2 mg/mL HAuCl_4_ and 120 µL 26.03 mg/L CO were added, diluted to 0.8 mL with water, and mixed well. After 5 min, 100 µL 0.1 mol/L trisodium citrate, 110 µL 0.01 mol/L HCl and 80 µL 1.0 mg/mL HAuCl_4_ were added. The tube was heated in a 75 °C water bath for 20 min, and cooled with ice–water. Then, 57 µL 20 µmol/L VBB and 30 µL 1 mol/L NaCl solutions were added, and diluted to 1.5 mL with water. The SERS at 1620 cm^−1^ (I), and the blank (I_0_) without of CO were recorded. The ΔI = I − I_0_ were calculated.

**The preparation of SEM samples:** According to the experimental methods, 1.5 mL reaction solution was taken into a centrifuge tube for 20 min (150 × 100 r/min). Then, the supernatant was removed and diluted to 1.5 mL, with 30 min ultrasonic oscillation. The latter was repeated twice and then 1.5 mL water was added; we then applied 2 μL samples to the silicon wafers for natural drying. After all the steps mentioned above, SEM was performed.

**The preparation of CO samples:** A total of 10 mL absorb liquid (pH 5.2 buffer + HAuCl_4_) in u-shaped tube, which connected to the atmospheric sampling instrument, sampling for 60 min with the flow velocity of 0.5 L/min, was used. Then, this was transferred into 5 mL colorimetric tubes. There were also some blank experiments being set for CO detection.

## 3. Results and Discussion

### 3.1. Analytical Principle

When a beam of light irradiates nanoparticles with molecules, it generates Rayleigh scattering and Raman scattering. Rayleigh scattering is an elastic scattering. Here, it was much greater than that of the Raman scattering intensity of inelastic scattering. That is to say, the inelastic scattering intensity can be ignored when compared with the Rayleigh scattering. When the molecules were adsorbed on the nanoparticle substrate that produced the SERS effect, the intensity was enhanced greatly. The total scattering intensity (I_S_) was the sum of RRS and SERS. That is, I_S_ = I_RRS_ + I_SERS_. In order to achieve this goal, we explored the CO system with SERS and RRS effects.

CO has strong reducibility in a pH 5.2 HAc-NaAc buffer. Therefore, it can react with HAuCl_4_ to generate red gold nanoparticles, and with the increase of CO concentration, more gold nanoparticles are generated, and it has SERS activity and RRS effect. The RRS and SERS signals (in the presence of the SERS probe VBB) have a positive correlation with the concentration of CO. According to this, two simple and rapid SERS and RRS methods can be established for the analysis of CO. Furthermore, the gold nanoparticles have rich free surface electrons that can enhanced the redox electron transfer and exhibit strong catalysis of the ethanol reduction of HAuCl_4_ to produce gold nanoparticles with SERS activity and RRS effects. It can be deduced from Equation 1 that under the catalysis of the gold nanoparticles (Au)_n_ generated in the first step, HAuCl_4_ is reduced to Au by ethanol, and finally autocatalytic growth occurs to AuNPs, and the corresponding ethanol is oxidized to acetic acid. Meanwhile, the formed small Au crystal nucleus also catalyzed the redox reaction of ethanol-HAuCl_4_, which is called autocatalysis. With the increase of CO concentration, the SERS, RRS and both sums increased linearly due to the gold nanoparticles increasing in the catalytic analytical system. Thus, a novel and highly sensitive SERS/RRS coupled di-mode method was established for the determination of trace CO (Figure 1).

Nanocatalytic reaction: (1)$${\mathrm{HAuCl}}_{4}+\mathrm{ethanol}\;\left(\mathrm{En}\right)\overset{(\mathrm{Au})n}{\underset{\mathrm{catalysis}}{\to }}\mathrm{AuNPs}+\mathrm{acetic}\;\mathrm{acid}$$

### 3.2. SERS Spectra

A normal Raman scattering signal is very weak, and SERS is very strong due to some organic molecules adsorbing on the surface of nanoparticles to produce the surface plasmon resonance scattering. In general, the SERS intensity responds to molecular probe concentrations. Recently, our group has found that the SERS intensity is also linear to the concentration of nanoparticles as a sole substrate when molecular probe concentrations are held constant. This has been used in SERS quantitative analysis. Using VBB as a molecular probe, the CO-HAuCl_4_ system exhibited three weak SERS peaks at 1173, 1400 and 1620 cm^−1^ (Figure 2A). The peaks at 1173, 1400 and 1620 cm^−1^ are related to -NH_2_ and C-H bending vibrations, and C=C and C=N stretching vibrations. The most sensitive peak at 1620 cm^−1^ was selected for use. In a 75 ℃ water bath, the reaction of HAuCl_4_-En is very slow. Using the AuNPs come from the CO-HAuCl_4_ reaction as catalyst, the reaction of HAuCl_4_-En can quickly catalyze the production of gold nanoparticles. With the increase of CO concentration, the AuNPs produced by the first reaction of CO-HAuCl_4_ increase, and the reaction of HAuCl_4_-En was catalyzed by AuNPs to generate a large number of gold nanoparticles, and the SERS signal enhanced greatly (Figure 2B), and the peak is enhanced linearly with CO concentration increasing. In addition, both HAuCl_4_-gluose and HAuCl_4_-citrate systems also exhibit a strong SERS peak at 1620 cm^−1^, and its intensity also increased with CO concentration increasing (Appendix A).

### 3.3. RRS Spectra

RRS is a kind of sensitive molecular spectral technology for the study of nanoparticles. For the CO-HAuCl_4_ system, CO reacted with HAuCl_4_ to generate gold nanoparticles with an RRS peak at 290 nm (Figure 3A). In Figure 3A, no obvious RRS peak was observed around 530 nm, mainly because the CO-HAuCl_4_ system produced few gold nanoparticles and the Rayleigh scattering effect were not obvious, and the RRS intensity linearly enhanced with the increase of CO concentration in the range of 173.5–12,147.3 ng/mL. For the CO-PdCl_2_ system, there are two obvious RRS peaks at 270 and 360 nm (Appendix A). The RRS intensity at 360 nm is linear to CO concentration in the range of 0.1–2.0 μg/mL CO. Nano-catalysis is a new way to enhance analytical signals. Comparing the sensitivity of Pd (II)-CO and Au (III)-CO systems, it was found that the Au (III)-CO system was more sensitive. To increase the RRS sensitivity for detection of CO, the Au (III)-CO system was coupled with nanocatalytic reaction. At pH 5.2, HAc-NaAc buffer, the CO-HAuCl_4_-En catalytic system exhibited three RRS peaks at 290, 370 and 535 nm. The RRS peaks at 290 and 370 nm belong to the maximum emission peak of the light source, while the peak at 535 nm is the RRS peak of the spherical AuNPs. Because the intensity of the incident light is large, the RRS signal at 370 nm is stronger than the signal at 535 nm and the regularity is also very good. When CO concentration increases, the peak at 370 nm increases linearly, and this was selected for use (Figure 3B). In addition, the gold nanoparticles from the CO-HAuCl_4_ reaction, also catalyzed the gold nano-reaction of HAuCl_4_-glucose. There were also three RRS peaks at 290, 370 nm and 535 nm (Appendix A). Similar to the reducer of En and glucose-containing alcohols, using glycerol as reducer, it also exhibited three RRS peaks at 290, 370 and 535 nm (Appendix A). As for the CO-HAuCl_4_-citate catalytic system, it has two RRS peaks at 370 and 550 nm (Appendix A). Out of the four catalytic systems, the En system is most sensitive and was chosen for detection of CO.

### 3.4. UV Absorption Spectra

CO could reduce HAuCl_4_ to generate AuNPs in the pH 5.2 buffer solution, and the color of the system changed from colorless to pale blue. It had a weak surface plasma resonance (SPR) absorption peak at 548 nm. Along with the increase of the CO concentration, the absorption peak at 548 nm intensity increased slowly (Figure 4A). The CO-HAuCl_4_-En catalytic system exhibited a strong absorption peak at 550 nm, which was ascribed to the SPR absorption peak of AuNPs. With the increase of the CO concentration, the peaks increased (Figure 4B), but their sensitivity was lower than the RRS method.

### 3.5. Gold Nano-Enzyme Catalysis

We known that Au^3+^ and En can adsorb on the surface of AuNPs from the CO-Au^3+^ reaction. The AuNPs have rich surface electrons that enhance the Au^3+^ -En redox electron transfer to form more gold nanoparticles (Figure 5). The Au^3+^ was reduced step-by-step to Au^+^ and Au; meanwhile, the En was oxidized step-by-step to acetaldehyde (AD) and acetic acid (AA). We also need to pay attention to the catalytic effect of the Au nuclei formed by the redox reaction. That is to say, the autocatalytic reaction also exists in this system. At the same time, although CO reduces Au^3+^ at a high speed, the reduction ability is weak, so the SERS and RRS sensitivity of the first reaction is very weak. After the nanogold seed is generated through the first-step reaction, the nanogold seed can be used as a catalyst, which greatly improves the growth efficiency of gold nanoparticles. The specific reaction mechanism is shown in Figure 5.

### 3.6. Scanning Electron Microscopy (SEM), Transmission Electron Microscope (TEM) and Energy Spectrum

As the figure shows (Figure 6A), the gold nanoparticles of the CO-HAuCl_4_ system have uniform particle sizes. In the TEM inset of Figure 6a, it can be seen that the average particle size of the gold nanoparticles is about 10 nm. The energy spectrum diagram of the nanogold system was obtained by using a transmission electron microscope with 200 kV voltage. The Au peaks appeared at 1.6, 2.1, 2.4 and 9.6 keV. The SEM diagram (TEM inset in Figure 6B) of HAuCl_4_-En was shown as in Figure 6b, with the progress of the catalytic reaction, Au (III)-En system produced more gold nanoparticles. They have four main spectral peaks at 1.6, 2.1, 2.4, and 9.6 keV, respectively, and the gold content is higher than that of the CO-Au (III) system.

### 3.7. Conditions Optimization

The two commonly used buffers including acetate and phosphate were considered, and the former was chosen in this study. As in Appendix A, the pH of the acetate buffer solution was 5.2, which was achieved by a large ΔI value, and was chosen for use. As shown in Appendix A, when the buffer solution concentration reached 0.24 mmoL/L, ΔI was largest, which was why the buffer solution concentration was chosen as 0.24 mmol/L in this experiment. The effect of HAuCl_4_ concentration on the system ΔI was also researched. When the concentration of HAuCl_4_ for the redox reaction was 13.3 μg/mL, ΔI was largest, so the concentration of 13.3 μg/mL HAuCl_4_ was chosen (Appendix A). The effect of HAuCl_4_ concentration of the nano-catalytic reaction on the ΔI was also examined. Results (Appendix A) show that when the concentration of HAuCl_4_ was 66.6 μg/mL, the ΔI was the largest, so the concentration of 66.6 μg/mL HAuCl_4_ was chosen. When the En concentration was 10.6% (v/v) in this experiment the ΔI could achieve the largest in the condition (Appendix A). The effect of reaction temperature and time on the ΔI were examined. Results (Appendix A) show that the tube was heated in a 75 °C water bath for 20 min, giving the highest ΔI value, and a reaction temperature of 75 °C for heating for 20 min was selected for use. NaCl was a good sensitizer for improvement of SERS signals due to aggregation of the nanoparticles, the effect of its concentration shows that 20 mmol/L gives a high intensity and was chosen for use (Appendix A). A 0.76 μmol/L VBB obtained the highest intensity and was selected for use (Appendix A).

### 3.8. Calibration Curve

The standard curves for SERS and RRS methods were obtained according to the experimental procedure. As for the system of CO-HAuCl_4_, the CO concentration had a good linear relationship with ΔI and ΔI_370 nm_ when it was in the range of 30–5230 ng/mL and 100–1500 ng/mL, respectively. The SERS method is more sensitive than the RRS. To obtain high sensitivity and accuracy, the catalytic SERS, RRS and SERS/RRS working curves for the En system were examined as in Table 1. It can be seen that the HAuCl_4_-En SERS/RRS di-mode method is the most sensitive and was chosen for use. The relative standard deviation of five determinations of 100 ng/mL CO by SERS, RRS and di-mode were 7.0%, 4.5%, and 5.6%, respectively. It was compared to the reported methods for CO [26,27,28,29,30,31,32,33,40,41,42,43,44] (Appendix A), and the SERS/RRS coupled di-mode is one of the most sensitive methods.

### 3.9. The Influence of Coexisting Substance

The interferences of common coexistence in the catalytic system were investigated according to the experimental procedure. The results (Appendix A) show that the relative error was within ±10%. In the CO-HAuCl_4_-En system, using acetate as the buffer solution, 10 µg/mL Ca^2+^, Zn^2+^, K^+^, BrO_3_^−^, Na_2_S, SeO_3_^2−^, Ni^2+^, Cr^3+^, glucose, ethanol and methyl alcohol; 8 µg/mL Co^2+^, Mg^2+^, Fe^3+^, Pb^2+^, Al^3+^, Mn^2+^, Na_2_S_2_O_3_ and formaldehyde has little effect on the determination of 0.1 µg/mL CO. The results show that this nano-catalytic SERS/RRS di-mode method had a good selectivity.

### 3.10. Analysis of Samples

When CO concentration was 667 ppm, about 50% oxygen hemoglobin changed into carbonyl hemoglobin. As we know, CO is produced by burning cigarettes. Therefore, we measured six samples, like indoor air, burning gas and cigarette combustion gas, etc. According to the detection methods, there was no CO detected in Samples 1–3 (Table 2), which was due to no burning of carbon material. After burning paper and cigarettes, CO was found in the collected combustion gases. This is due to the incomplete burning of paper and cigarettes, which released the CO that can be detected by the di-mode method (Sample 4–6) with a relative standard deviation (RSD) of 1.8–4.2%. The di-mode results were in agreement with that of CO-Pd(II) spectrophotometry which is a standard method. This indicates that this new method is accurate and reliable.

## 4. Conclusions

CO has strong reducibility in pH 5.2 HAc-NaAc buffer solution. So it can react with Au(III), generating Au nanoparticles, which exhibit SERS activity and RRS effects. With the increase of CO concentration, both scattering values also increased. Accordingly, two simple and rapid SERS and RRS methods for CO could be established, respectively. Based on the Au nanoparticle catalysis of HAuCl_4_-En to amplify the SERS and RRS signals, a new sensitive SERS/RRS coupled di-mode method was established for testing trace CO. This SERS/RRS coupled di-mode method was more sensitive than the two SERS and RRS single mode methods.

## Figures and Tables

**Figure 1 nanomaterials-10-00450-f001:**
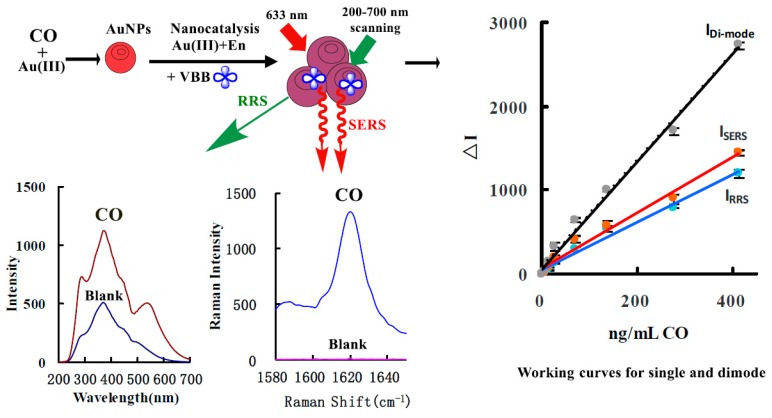
Analytical principle of SERS/RRS coupled di-mode determination of CO by nanogold catalytic amplification.

**Figure 2 nanomaterials-10-00450-f002:**
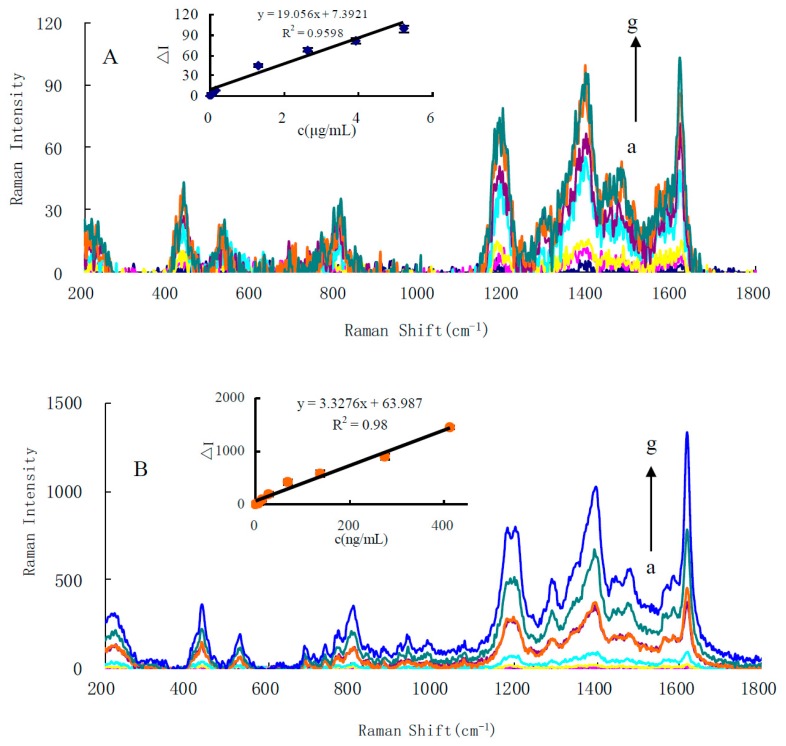
SERS spectra of (**A**) CO-HAuCl_4_-VBB and (**B**) CO-HAuCl_4_-En-VBB systems. (**A**), a: 0.24 mmol/L pH 5.2 HAc-NaAc + 50.2 µg/mL HAuCl_4_ + 0.76 µmol/L VBB + 5.9 mmol/L NaCl; b: a + 0.03 µg/mL CO; c: a + 0.13 µg/mL CO; d: a + 0.31 µg/mL CO; e: a + 2.62 µg/mL CO; f: a + 3.92 µg/mL CO; g: a + 5.23 µg/mL CO. (**B**), a: 0.24 mmol/L pH 5.2 HAc-NaAc + 10.6% En + 0.58 mmol/L HCl + 52.9 µg/mL HAuCl_4_ + 0.76 µmol/L VBB + 5.9 mmol/L NaCl; b: a + 13.8 ng/mL CO; c: a + 27.5 ng/mL CO; d: a + 68.9 ng/mL CO; e: a + 137.7 ng/mL CO; f: a + 275.4 ng/mL CO; g: a + 413.2 ng/mL CO.

**Figure 3 nanomaterials-10-00450-f003:**
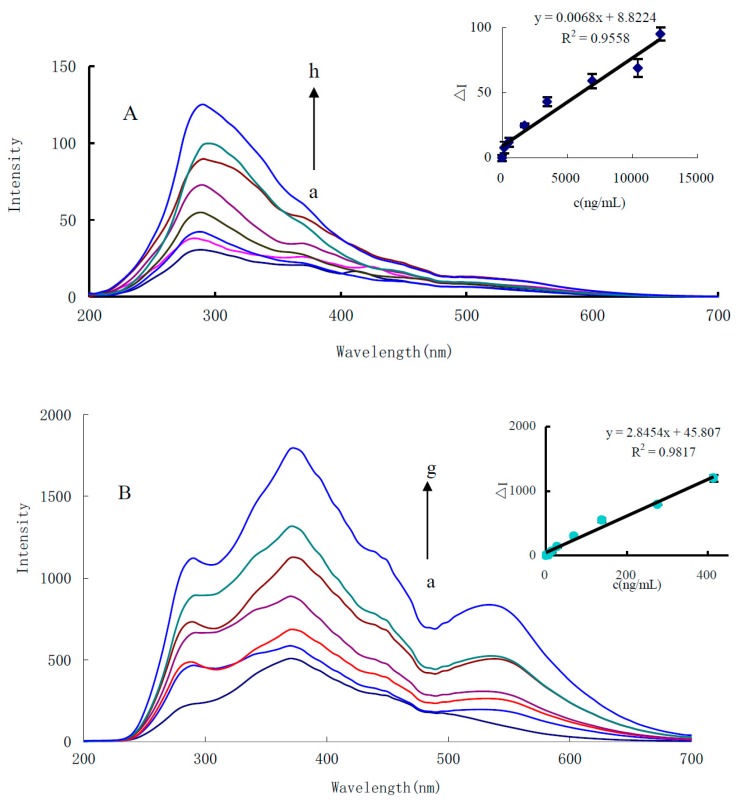
RRS spectra of (**A**) CO-HAuCl_4_ system and (**B**) CO-HAuCl_4_-En catalytic system. (**A**) a: pH 5.2 HAc-NaAc +13.3 µg/mL HAuCl_4_; b: a + 173.5 ng/mL CO; c: a + 520.6 ng/mL CO; d: a + 1735.3 ng/mL CO; e: a + 3470.7 ng/mL CO; f: a + 6941.3 ng/mL CO; g: a + 10412.0 ng/mL CO; h: a + 12147.3 ng/mL CO. (**B**) a: pH 5.2 HAc-NaAc + 10.6% En + 0.58 mmol/L HCl + 66.6 µg/mL HAuCl_4_; b: a + 13.8 ng/mL CO; c: a + 27.5 ng/mL CO; d: a + 68.9 ng/mL CO; e: a + 137.7 ng/mL CO; f: a + 275.4 ng/mL CO; g: a + 413.2 ng/mL CO.

**Figure 4 nanomaterials-10-00450-f004:**
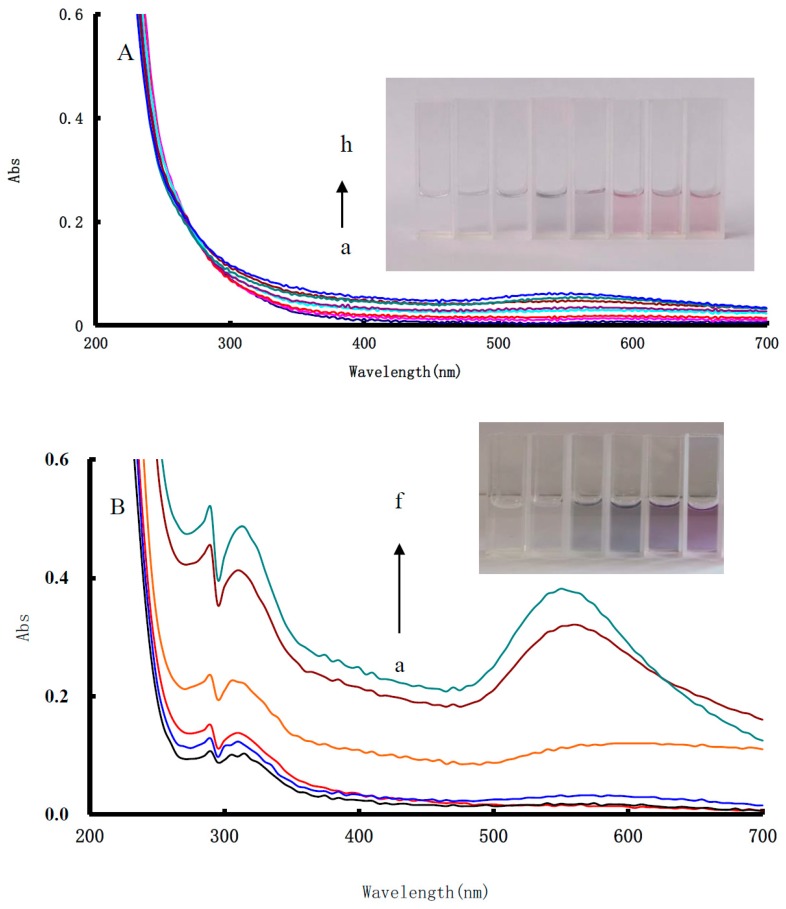
Absorption spectra of CO-HAuCl_4_ system (**A**) and CO-HAuCl_4_-En catalytic system (**B**). (**A**) a: pH 5.2 HAc-NaAc + 13.3 μg/mL HAuCl_4_; b: a + 0.25 μg/mL CO; c: a + 0.5 μg/mL CO; d: a + 1.0 μg/mL CO; e: a + 1.5 μg/mL CO; f: a + 2.0 μg/mL CO; g: a + 2.5 μg/mL CO; h: a + 3.125 μg/mL CO. (**B**) a: 0.24 mmol/L pH 5.2 HAc-NaAc +10.6% En + 0.58 mmol/L HCl + 66.6 µg/mL HAuCl_4_; b: a + 0.01 µg/mL CO; c: a + 0.03 µg/mL CO; d: a + 0.07 µg/mL CO; e: a + 0.41 µg/mL CO; f: a + 0.69 µg/mL CO.

**Figure 5 nanomaterials-10-00450-f005:**
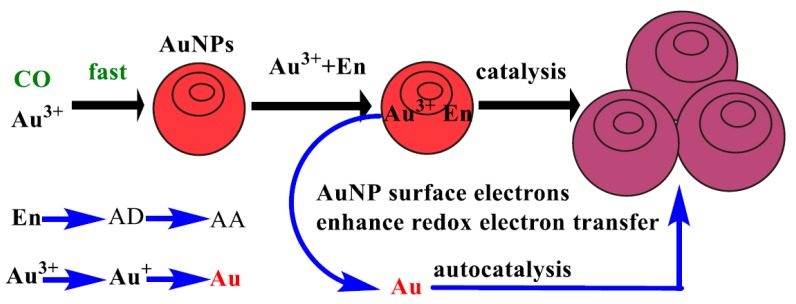
Mechanism of the AuNP catalysis of En reduction of Au^3+^.

**Figure 6 nanomaterials-10-00450-f006:**
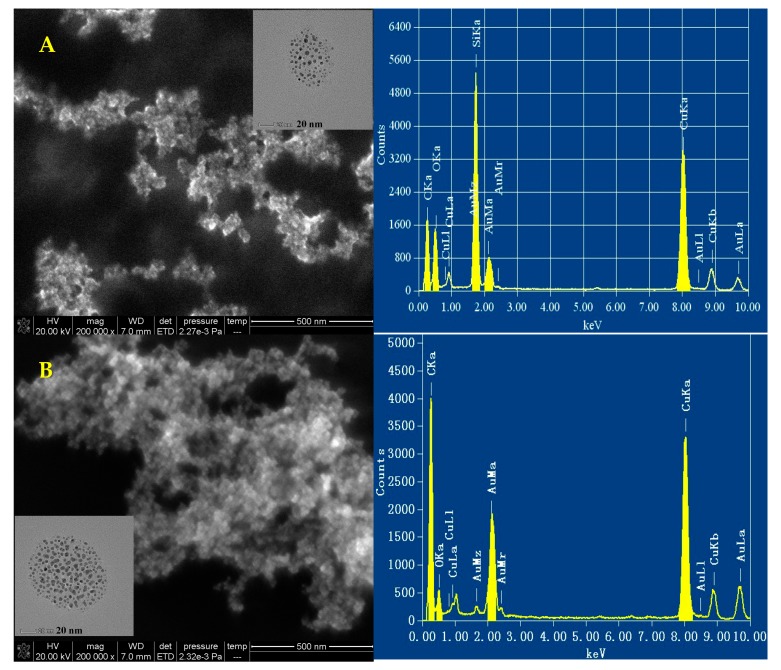
The SEM and energy spectral pictures. (**A**) pH 5.2 buffer + 13.3 μg/mL HAuCl_4_ + 1.0 μg/mL CO; (**B**) pH 5.2 buffer + 66.6 μg/mL HAuCl_4_ + 1.0 μg/mL CO + 10.6% En.

**Table 1 nanomaterials-10-00450-t001:** Analysis features of RRS and absorption methods for CO.

System	Methods	Regression Equation	LR (ng/mL)	Coefficient	DL (ng/mL)
HAuCl_4_	SERS	ΔI = 0.019C + 7.4	30–5230	0.9598	20 ± 0.04
	RRS	ΔI = 0.0068C + 8.8	100–1500	0.9558	80 ± 0.13
HAuCl_4_-En	SERS	ΔI = 3.33C + 63.9	5.0–413	0.9800	3 ± 0.02
	RRS	ΔI = 2.84C + 45.8	10–413	0.9817	6 ± 0.05
	Di-mode	ΔI = 6.33C + 62	3.0–413	0.9851	1 ± 0.02

LR: linear range; DL: detection limit.

**Table 2 nanomaterials-10-00450-t002:** Sample analysis results.

Samples	Sampling Temperature (°C)	Measured Value (mg/m^3^)	Average (mg/m^3^)	RSD (%)	Ref. Results (mg/m^3^)
1	18	Not detected	Not detected	-	Not detected
2	20	Not detected	Not detected	-	Not detected
3	18	Not detected	Not detected	-	Not detected
4	20	0.0076 0.0074 0.0073 0.0075 0.0073	0.00752	1.8	0.00810
5	20	0.0039 0.004 0.0038 0.0039 0.0041	0.00394	2.6	0.0035
6	18	0.0011 0.0012 0.0012 0.0011 0.0012	0.00116	4.2	0.00120

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
