# Peer review of "A Highly Sensitive SERS and RRS Coupled Di-Mode Method for CO Detection Using Nanogolds as Catalysts and Bifunctional Probes"

_nanomaterials, 2020, doi:10.3390/nano10030450_

Round 1

Reviewer 1 Report

The manuscript concerns the detection of CO in very low concentrations (in the range of tens and hundreds of ng/mL depending on the employed technique) with a linear response in the whole range (30-5230 ng/mL and 100-1500 ng/mL). The detection is based on a chemical reduction of Au(III) induced by CO which, consequently, leads to Au nanoparticles formation. Two sensitive methods are used for the evaluation of the extent of nanoparticles formation and the CO detection limit determination: (i) Surface-Enhanced Raman scattering (SERS) exploiting a model dye (Vitoria Blue B) and (ii) resonance Rayleigh scattering (RRS).

Although the data are brilliant, there are several important issues which have to be addressed before publication. Therefore major revisions are required.

The data should be discussed in more details. Currently, there are virtually only experimental data without any further discussion of them. Especially, parts 3.1, 3.2, 3.3., 3.5, 3.6, 3.9, 3.10 should be enlarged, the data critically discussed. Similarly, the experimental part should be substantially enlarged and more detailed. In the current form, it is not possible to repeat many procedures. For instance, it is not clear how to prepare the samples described on page 3, line 132: “then diluted it to 1.5mL”. According to the previous sentences, the system has already approx.. 5 mL. The same situation is on page 3, line 137. What are the individual peaks in RRS stemming from? I mean, 290 nm and 370 nm? The peaks at 535 and/or 550 nm in RRS are attributed to Au nanoparticles. SEM data are not convincing as for the size of Au nanoparticles. Indeed, in order to discuss the diameters of particles around 10 nm and/or 15 nm, a higher magnification should be used and images given with the maximal scale of 50 nm (definitely not 500 nm as it is in the current version). The size of nanoparticles is crucial as for the scattering intensity: the bigger the nanoparticles formed, the more intense scattering. Therefore, any critical evaluation and another independent technique should be used as for particles size determination. The authors should give evidence that a higher amount of nanoparticles is produced in CO-HAuCl4-En systems in comparison to CO-HAuCl4. They deduce it from SERS spectra, however, the SERS spectra are more complex. Moreover, for an unambiguous detection of any species at least three peaks in SERS should be evaluated and not only one peak. The relative intensity of peaks is related to the form of species being adsorbed on nanoparticle surface. There is no comparison with the detection limits of CO-Pd(II) spectrophotometry as a standard method (referencing is missing) in Table S1.

Furthermore, although I appreciate the chapters dealing with the verification of the influence of coexisting species in tested samples, as well as, analysis of real samples (e.g. burning gas, cigarette combustion gas), the description of the sample preparation should be given in experimental part.

Several parts in the main text should be given in experimental part: lines 244-248, page 8; 302-306, page 10

English should be corrected on many places in the main text (lines 159, 173, 177, 179, 200, 224, 249, 291-293, etc.), in the title “for CO detection”.

Also the sentence on page 2, line 89 is not clear: “haemoglobin (Hb) and haemoglobin (HbCO) by combined the dissolved CO”. Possibly it should have been free haemoglobin (Hb) and haemoglobin with CO (HbCO).

The labelling of the systems is inconsistent: HAC-NaAc-CO-HAuCl4 and/or CO-HAuCl4 labels in the main text and in figures. It should be unified and each label introduces in experimental part.

In Figure 1, the graph presenting working curves for single and di-mode: y axis is not described and error bars are not shown. Similarly, error bars are not presented in insets of Figures 2, 3.

In Figure caption of Figure 2B, VBB is missing in the label (should be CO-HAuCl4-En-VBB so that it is consistent with 2A description). Although “VBB “ should be there taking into account the spectra presented and system description provided in figure caption.

The equations on page 4, lines 163-165 are not discussed in the text. By the way, the first equation is not correct (6H+ should be in products).

Page 4, line 177, Figure 2A instead of 1A and line 179, Figure 2B instead of 1B should have been referenced.

In Table 1, abbreviations LR, DL are not explained at all.

Detailed description of HAuCl4-glucose and HAuCl4-citrate systems, mentioned on page 5, line 180-181, are not given. Also, the preparation of HAc-NaAc-CO-PdCl2 system mentioned on page 5, line 196, is not described.

Figure 5: “AuNP surface electrons were enhanced the redox”??

Lines 296-298, page 10 are redundant (already mentioned in Introduction)

Page 9, line 261: “as shown in Figure S6b, when a buffer solution concentration achieved 40 mmol/L”, however, there is 20 mmol/L as the maximal value in Figure S6b.

Author Response

Response to Reviewer 1 Comments

Comments and Suggestions for Authors

The manuscript concerns the detection of CO in very low concentrations (in the range of tens and hundreds of ng/mL depending on the employed technique) with a linear response in the whole range (30-5230 ng/mL and 100-1500 ng/mL). The detection is based on a chemical reduction of Au(III) induced by CO which, consequently, leads to Au nanoparticles formation. Two sensitive methods are used for the evaluation of the extent of nanoparticles formation and the CO detection limit determination: (i) Surface-Enhanced Raman scattering (SERS) exploiting a model dye (Vitoria Blue B) and (ii) resonance Rayleigh scattering (RRS).

Although the data are brilliant, there are several important issues which have to be addressed before publication. Therefore major revisions are required.

Point 1: The data should be discussed in more details. Currently, there are virtually only experimental data without any further discussion of them. Especially, parts 3.1, 3.2, 3.3., 3.5, 3.6, 3.9, 3.10 should be enlarged, the data critically discussed. Similarly, the experimental part should be substantially enlarged and more detailed. In the current form, it is not possible to repeat many procedures. For instance, it is not clear how to prepare the samples described on page 3, line 132: “then diluted it to 1.5mL”. According to the previous sentences, the system has already approx.. 5 mL. The same situation is on page 3, line 137. What are the individual peaks in RRS stemming from? I mean, 290 nm and 370 nm? The peaks at 535 and/or 550 nm in RRS are attributed to Au nanoparticles. SEM data are not convincing as for the size of Au nanoparticles. Indeed, in order to discuss the diameters of particles around 10 nm and/or 15 nm, a higher magnification should be used and images given with the maximal scale of 50 nm (definitely not 500 nm as it is in the current version). The size of nanoparticles is crucial as for the scattering intensity: the bigger the nanoparticles formed, the more intense scattering. Therefore, any critical evaluation and another independent technique should be used as for particles size determination. The authors should give evidence that a higher amount of nanoparticles is produced in CO-HAuCl4-En systems in comparison to CO-HAuCl4. They deduce it from SERS spectra, however, the SERS spectra are more complex. Moreover, for an unambiguous detection of any species at least three peaks in SERS should be evaluated and not only one peak. The relative intensity of peaks is related to the form of species being adsorbed on nanoparticle surface. There is no comparison with the detection limits of CO-Pd(II) spectrophotometry as a standard method (referencing is missing) in Table S1.

Furthermore, although I appreciate the chapters dealing with the verification of the influence of coexisting species in tested samples, as well as, analysis of real samples (e.g. burning gas, cigarette combustion gas), the description of the sample preparation should be given in experimental part.

Response 1: Following your suggestion, we have discussed in depth in parts 3.1, 3.2, 3.3., 3.5, 3.6, 3.9, 3.10.

We have substantially enlarged and detailed the experimental part. For the reviewer's question, “For instance, it is not clear how to prepare the samples described on page 3, line 132: “then diluted it to 1.5mL”. According to the previous sentences, the system has already approx.. 5 mL.” may be that our description is unclear, we have modified it, but the total volume is less than 1.5 mL. (200 µL HAc-NaAc + 100 µL HAuCl4 + 120 µL CO + 57 µL VBB + 30 µL NaCl = 507 µL = 0.507 mL). "On page 3, line 137", we have also modified the representation of the method.

The RRS peaks at 290 and 370 nm belong to the maximum emission peak of the light source, while the peak at 535 nm is the RRS peak of the spherical AuNPs. Because the intensity of the incident light is large, the RRS signal at 370 nm is stronger than the signal at 535 nm and the regularity is also very good.

We supplemented the morphological characterization of the nanoparticles with a transmission electron microscope (inset in Figure 6a), a higher magnification has been used and image given with the maximal scale of 20 nm. And we can see that they have better dispersibility, with an average particle size of about 10 nm.

From the UV absorption spectra in section 3.4 and its corresponding solution color, it can be seen that under the same CO concentration, the absorption peak of the CO-HAuCl4-En system is stronger and the solution color is darker. At the same time, the SEM and energy spectrum of Section 3.6 also show that the gold content in the CO-HAuCl4-En system increase.

Regarding the reviewer's opinion that " Moreover, for an unambiguous detection of any species at least three peaks in SERS should be evaluated and not only one peak", thank you very much for your suggestion, but we consider that it is very difficult to achieve a linear relationship between the intensity of the three SERS peaks and the concentration of CO simultaneously. Therefore, only the most sensitive SERS peaks were selected for quantitative analysis.

The detection limit of CO-Pd (II) spectrophotometry as a standard method (Reference 44) has been added in Table S1 for comparison.

The description of the sample preparation has been transferred to the experimental part (2.3 Experiment methods).

Point 2: Several parts in the main text should be given in experimental part: lines 244-248, page 8; 302-306, page 10

Response 2: Several parts of the text have been given in the experimental part (2.3 Experiment methods): lines 244-248, page 8; 302-306, page 10.

Point 3: English should be corrected on many places in the main text (lines 159, 173, 177, 179, 200, 224, 249, 291-293, etc.), in the title “for CO detection”.

Response 3: We have corrected many places in the text (now lines 197, 211, 215, 222, 242, 269, 293, 339-341, etc.), "for CO detection" in the title.

Point 4: Also the sentence on page 2, line 89 is not clear: “haemoglobin (Hb) and haemoglobin (HbCO) by combined the dissolved CO”. Possibly it should have been free haemoglobin (Hb) and haemoglobin with CO (HbCO).

Response 4: The sentence on page 2, line 89 (now line 93) has been modified to “Chen et al. [33] used bovine hemoglobin with a high affinity for CO to combine dissolved CO in water to form a two-component system consisting of hemoglobin (Hb) and carboxyhemoglobin (HbCO).”

Point 5: The labelling of the systems is inconsistent: HAC-NaAc-CO-HAuCl4 and/or CO-HAuCl4 labels in the main text and in figures. It should be unified and each label introduces in experimental part.

Response 5: The labelling of the systems in the main text and the figures has been modified to a unified format: CO-HAuCl4.

Point 6: In Figure 1, the graph presenting working curves for single and di-mode: y axis is not described and error bars are not shown. Similarly, error bars are not presented in insets of Figures 2, 3.

Response 6: Thank you for your reminder, we have added the description of the y-axis (ΔI) to the working curve of Figure 1, and added the error bar. Error bars have also been added to the insets of Figures 2 and 3.

Point 7: In Figure caption of Figure 2B, VBB is missing in the label (should be CO-HAuCl4-En-VBB so that it is consistent with 2A description). Although “VBB “ should be there taking into account the spectra presented and system description provided in figure caption.

Response 7: In the Figure caption of Figure 2B, VBB has been added.

Point 8: The equations on page 4, lines 163-165 are not discussed in the text. By the way, the first equation is not correct (6H+ should be in products).

Response 8: According to the comment of the 4th reviewer, we have deleted the two equations on lines 163-164 and added a discussion of the equation on line 165 (now lines 203) in the text.

Point 9: Page 4, line 177, Figure 2A instead of 1A and line 179, Figure 2B instead of 1B should have been referenced.

Response 9: In line 177 (now line 215) on page 4, Figure 1A has been modified to Figure 2A, and Figure 1B in line 179 (now line 222) has been modified to Figure 2B.

Point 10: In Table 1, abbreviations LR, DL are not explained at all.

Response 10: The abbreviations LR and DL have been explained below Table 1. LR and DL are abbreviations for linear range and detection limit, respectively.

Point 11: Detailed description of HAuCl4-glucose and HAuCl4-citrate systems, mentioned on page 5, line 180-181, are not given. Also, the preparation of HAc-NaAc-CO-PdCl2 system mentioned on page 5, line 196, is not described.

Response 11: A detailed description of the preparation of the HAuCl4-glucose system, HAuCl4-citrate system mentioned on page 5, line 180-181 (now line 223), and the HAc-NaAc-CO-PdCl2 system mentioned on page 5, line 196 (now line 239) is given in the section of 2.3Experiment methods.

Point 12: Figure 5: “AuNP surface electrons were enhanced the redox”??

Response 12:"AuNP surface electrons were enhanced the redox" has been modified to "AuNP surface electrons enhance redox electron transfer".

Point 13: Lines 296-298, page 10 are redundant (already mentioned in Introduction)

Response 13: Lines 296-298 on page 10 have been deleted.

Point 14: Page 9, line 261: “as shown in Figure S6b, when a buffer solution concentration achieved 40 mmol/L”, however, there is 20 mmol/L as the maximal value in Figure S6b.

Response 14: Page 9, line 261 (now line 309): “as shown in Figure S6b, when a buffer solution concentration achieved 40 mmol/L”, this sentence has been modified to "And as shown in Figure S6b, when the buffer solution concentration achieved 0.24 mmoL/L ".

Reviewer 2 Report

Reviewer #1: The authors describe the development of SERS and RRS coupled ci-mode method for the detection CO using in situ synthesis of gold nanoparticle and signal enhancement using ethanol precursor. Bifunctional property of gold nanoparticle is the major impact of the manuscript. There are several issues which must be addressed.

Major issues:
The title of the submitted manuscript is not sufficient to contain the aim of the research. The proposed method, SERS and RRS coupled di-mode, is developed for the detection of CO. However, the title is not containing the aim of the developed method.

In the experimental section (line 117), the author mentioned that the TEM was used to investigate the synthesized gold nanoparticle. However, the obtained morphology images of the gold nanoparticle in the Figure 6 (line 243) were analyzed by SEM. Author have to check the instrument they used and correct the manuscript appropriately.

In line 179, “`the SERS signal enhanced greatly (Figure 1B).” is not correct because in the Figure 1, there is not appeared A and B.

Compare the obtained results (ex. LOD or detection time) with previously reported paper as summarized table to emphasize the developed method.

General comments:
Extensive English proofreading is needed. There are many grammatical errors, incorrect sentence construction, etc.

Author Response

Response to Reviewer 2 Comments

Comments and Suggestions for Authors

Reviewer #1: The authors describe the development of SERS and RRS coupled ci-mode method for the detection CO using in situ synthesis of gold nanoparticle and signal enhancement using ethanol precursor. Bifunctional property of gold nanoparticle is the major impact of the manuscript. There are several issues which must be addressed.

Major issues:

Point 1: The title of the submitted manuscript is not sufficient to contain the aim of the research. The proposed method, SERS and RRS coupled di-mode, is developed for the detection of CO. However, the title is not containing the aim of the developed method.

Response 1:  The title has been modified to “A highly sensitive SERS and RRS coupled di-mode method for CO detection using nanogolds as catalyst and bifunctional probes”.

Point 2: In the experimental section (line 117), the author mentioned that the TEM was used to investigate the synthesized gold nanoparticle. However, the obtained morphology images of the gold nanoparticle in the Figure 6 (line 243) were analyzed by SEM. Author have to check the instrument they used and correct the manuscript appropriately.

Response 2:  In the experimental section (now line 120), the model and manufacturer of the SEM (S-4800 scanning electron microscope) have been added.

Point 3: In line 179, “`the SERS signal enhanced greatly (Figure 1B).” is not correct because in the Figure 1, there is not appeared A and B.

Response 3:  In line 179 (now line 222), the sentence “`the SERS signal enhanced greatly (Figure 1B).” has been modified to “the SERS signal enhanced greatly (Figure 2B).”

Point 4: Compare the obtained results (ex. LOD or detection time) with previously reported paper as summarized table to emphasize the developed method.

Response 4:  In Table 1S, we have compared the results (such as LOD) obtained with previously reported papers as summary table to emphasize the developed method.

Point 5: General comments:

Extensive English proofreading is needed. There are many grammatical errors, incorrect sentence construction, etc.

Response 5:  We have proofread the English of the thesis, and modified grammatical errors and incorrect sentence structure.

Reviewer 3 Report

This study belongs to a series of the authors’ group works relevant to the article title. The presented chemical data based on SERS/RRS are of use for CO sensing community. This paper is well represented but the following English should be corrected.

.

L-173: In general, the SERS intensity is respond to molecular probe concentration.(responds)

L-180:the peaks enhanced linearly with CO .(is enhanced).

L-301: , there is also CO can be generated by cigarettes. (?)

L-306: According the detection methods, (According to)

L:308: While after burned the paper and cigarette, CO was found in the space, (?)

Author Response

Response to Reviewer 3 Comments

Comments and Suggestions for Authors

This study belongs to a series of the authors’ group works relevant to the article title. The presented chemical data based on SERS/RRS are of use for CO sensing community. This paper is well represented but the following English should be corrected.

Point 1: L-173: In general, the SERS intensity is respond to molecular probe concentration.(responds)

Response 1:  The sentence of L-173 (now line 211) has been modified to “In general, the SERS intensity is responds to molecular probe concentration”.

Point 2: L-180:the peaks enhanced linearly with CO .(is enhanced).

Response 2:The sentence of L-180 (now line 222) has been modified to “the peak is enhanced linearly with CO concentration increasing.”

Point 3: L-301: , there is also CO can be generated by cigarettes. (?)

Response 3:The sentence of L-301 (now line 345) has been modified to “the gas also contains CO that burned cigarettes.”

Point 4: L-306: According the detection methods, (According to)

Response 4:The sentence of L-306 (now line 347) has been modified to “According to the detection methods,”

Point 5: L:308: While after burned the paper and cigarette, CO was found in the space, (?)

Response 5:The sentence of L-308 (now line 348) has been modified to “After burning paper and cigarettes, CO was found in the collected combustion gases.”

Reviewer 4 Report

The paper could be of interest but it requires major revisions. First of all, the English form and grammar must be revised by a native speaker. Some sentences are hard to understand. In the Introduction, the physical basis of RRS should be better explained, since it is not a common technique as SERS it is. In the Results and Discussion, after the first paragraph, there are two unbalanced reaction equations that are not commented in the text and they should be, or should be deleted. The analytical method could be very interesting but it must be well characterized. In particular, the gold nanoparticles seem to be really aggregate from the SEM pictures reported in Figure 6. In the text, authors claim for 10 nm particles but it really hard to believe that since the bar scale is 500 nm and the resolution of the images is not very high. How does it change the average size of the nano-particles from batch to batch. This is crucial for a CO monitoring system. Moreover, the SERS enhancing factor should be better quantified. All the quantities that are measured or estimated by experiments must be reported with their errors otherwise they are meaningless. Please, add all the equations of the linear regressions in the inset of the plot and not in the text. Please, quantify the limit of detection and the sensitivities with the errors for each system. Please, try to explain the physical and chemical mechanisms that are at the basis of the different response of the two nano-complexes. 

Author Response

Response to Reviewer 4 Comments

Comments and Suggestions for Authors

Point 1: The paper could be of interest but it requires major revisions. First of all, the English form and grammar must be revised by a native speaker. Some sentences are hard to understand. In the Introduction, the physical basis of RRS should be better explained, since it is not a common technique as SERS it is. In the Results and Discussion, after the first paragraph, there are two unbalanced reaction equations that are not commented in the text and they should be, or should be deleted. The analytical method could be very interesting but it must be well characterized. In particular, the gold nanoparticles seem to be really aggregate from the SEM pictures reported in Figure 6. In the text, authors claim for 10 nm particles but it really hard to believe that since the bar scale is 500 nm and the resolution of the images is not very high. How does it change the average size of the nano-particles from batch to batch. This is crucial for a CO monitoring system. Moreover, the SERS enhancing factor should be better quantified. All the quantities that are measured or estimated by experiments must be reported with their errors otherwise they are meaningless. Please, add all the equations of the linear regressions in the inset of the plot and not in the text. Please, quantify the limit of detection and the sensitivities with the errors for each system. Please, try to explain the physical and chemical mechanisms that are at the basis of the different response of the two nano-complexes.

Response 1: English forms and grammar have been modified by native English speakers.

    In the second paragraph of the Introduction, we have explained the physical basis of RRS.

    After the first paragraph of Results and Discussion, two unbalanced reaction equations have been removed.

    It can be seen from Fig. 6 that the aggregation of the nanoparticles may be caused by centrifugation during the process. We supplemented the morphological characterization of the nanoparticles with a transmission electron microscope (inset in Figure 6a), and we can see that they have better dispersibility, with an average particle size of about 10 nm.

    And because our purpose is to establish a new method, rather than report a new material, we have not considered the calculation of the SERS enhancement factor for the time being. Thank you very much for your suggestions, we will consider adopting your suggestions in the future work.

    All equations for linear regression have been added to the inset of the figure.

    In Table 1, the error of the detection limit of each system has been added.

We have explained the physical and chemical mechanisms based on the different responses of two nanocomplexes in the section of 3.5 Gold nanoenzyme catalysis.

Reviewer 5 Report

Authors have submitted a manuscript describing a AuNPs-based sensor for CO detection.

Even if the idea to employ SERS and RRS techniques for the detection of CO is interesting, the approach of the Author is puzzling. Indeed, they employ CO for the synthesis of the nanoplatform that they subsequently analyze. It is hard to view some spot of potential application of this approach. On the other hand, if this is just a proof-of-concept, there are concerns regarding the control on the nanoplatform formation and how the CO should be recovered from atmosphere to be analyzed.

Author Response

Response to Reviewer 5 Comments

Comments and Suggestions for Authors

Authors have submitted a manuscript describing a AuNPs-based sensor for CO detection.

Point 1: Even if the idea to employ SERS and RRS techniques for the detection of CO is interesting, the approach of the Author is puzzling. Indeed, they employ CO for the synthesis of the nanoplatform that they subsequently analyze. It is hard to view some spot of potential application of this approach. On the other hand, if this is just a proof-of-concept, there are concerns regarding the control on the nanoplatform formation and how the CO should be recovered from atmosphere to be analyzed.

Response 1:  In our method, the nano-gold generated in the first step exists as a seed, and the nano-gold seed is used as a catalyst to greatly improve the sensitivity of the method. This method focuses on proposing a new concept. First, we obtain accurate CO concentration to control the growth of nano-gold seeds through reports in related literature. Secondly, the CO content in the air is very low, and we are all conducting experiments in a fume hood, so the issue of CO recovery is not considered.

Round 2

Reviewer 1 Report

I appreciate that the authors took into account many suggestions and comments I had. However, there are still a few points which should be improved, before publication in Nanomaterials, as it is listed below:

Page 9 line 290: instead of „figure below“ it should be Figure 5.

Page 9, Figure 6a: TEM image - numbers next to the scale bar should be more visible (i.e. in black colour)

Page 10, Figure 6b: TEM image for the system HAuCl4-En is required. It can be also included as an inset.

Page 10, lines 309-311, the sentence still does not make sense to me:“And as shown in Figure S6b, when the buffer solution concentration achieved 0.24 mmoL/L, ΔI was largest, which was why the buffer solution concentration was chosen as 40 mmol/L in this experiment.“ Why the buffer concentration of 0.24 mmol/L was not chosen and 40 mmol/L was chosen instead? The selected concentration seems to be absolutely outside of the tested range.

Author Response

Response to Reviewer 1 Comments

Comments and Suggestions for Authors

I appreciate that the authors took into account many suggestions and comments I had. However, there are still a few points which should be improved, before publication in Nanomaterials, as it is listed below:

Point 1: Page 9 line 290: instead of „figure below“ it should be Figure 5.

Response 1: Page 9 line 290 (now line 289): "figure below" has been modified to "Figure 5".

Point 2: Page 9, Figure 6a: TEM image - numbers next to the scale bar should be more visible (i.e. in black colour)

Response 2: Page 9, Figure 6a: TEM image-The number next to the scale bar has been added with a larger black font size.

Point 3: Page 10, Figure 6b: TEM image for the system HAuCl4-En is required. It can be also included as an inset.

Response 3: Page 10, Figure 6b: TEM image for the system HAuCl4-En has been added as an inset to Figure 6b.

Point 4: Page 10, lines 309-311, the sentence still does not make sense to me:“And as shown in Figure S6b, when the buffer solution concentration achieved 0.24 mmoL/L, ΔI was largest, which was why the buffer solution concentration was chosen as 40 mmol/L in this experiment.“ Why the buffer concentration of 0.24 mmol/L was not chosen and 40 mmol/L was chosen instead? The selected concentration seems to be absolutely outside of the tested range.

Response 4: Thank you for your reminder, this sentence has been modified to "And as shown in Figure S6b, when the buffer solution concentration achieved 0.24 mmol/L, ΔI was largest, which was why the buffer solution concentration was chosen as 0.24 mmol/L in this experiment”.

Reviewer 2 Report

The revised manuscript is suitable for the publication in this journal.

Author Response

Thank you.

Reviewer 4 Report

The paper could be published in the revised form

Author Response

Thank you.

Reviewer 5 Report

I regret to notice that the answer of the Authors to my considerations is just a no-answer. From my point of view, this manuscript has serious conceptual lacks.

Author Response

Response to Reviewer 5 Comments

Comments and Suggestions for Authors

I regret to notice that the answer of the Authors to my considerations is just a no-answer. From my point of view, this manuscript has serious conceptual lacks.

Response: Thank you for your suggestion, we will pay more attention to the definition of concepts in future work.